# *Parkia platycephala* Pods Modulate *Eimeria* spp. Parasite Load and Enhance Productive Performance in Naturally Infected Lambs

**DOI:** 10.3390/ani15192896

**Published:** 2025-10-03

**Authors:** Thalia Caldas da Silva, Gabrielle de Melo Oliveira, Osmar Macêdo Fortaleza Neto, Maycon Rodrigo de Souza Diniz, Joana Kellany Gonçalves de Andrade, José Gracione do Nascimento Souza Filho, Janaína Marques do Nascimento, Sara Silva Reis, Michelle de Oliveira Maia Parente, Arlan Araújo Rodrigues, Anderson de Moura Zanine, Henrique Nunes Parente, Ivo Alexandre Leme da Cunha

**Affiliations:** 1Science Center of Chapadinha, Federal University of Maranhão, Chapadinha 65.500-000, Brazil; thalia.cs@discente.ufma.br (T.C.d.S.); melo.gabrielle@discente.ufma.br (G.d.M.O.); osmar.fortaleza@discente.ufma.br (O.M.F.N.); rodrigo.maycon@discente.ufma.br (M.R.d.S.D.); joana.kellany@discente.ufma.br (J.K.G.d.A.); jgns.filho@discente.ufma.br (J.G.d.N.S.F.); janaina.marques@discente.ufma.br (J.M.d.N.); anderson.zanine@ufma.br (A.d.M.Z.); henrique.parente@ufma.br (H.N.P.); 2Postgraduate Program in Animal Science, Federal University of Paraíba, Areia 58.397-000, Brazil; sara.reis@academico.ufpb.br; 3Department of Animal Science, Federal University of Piauí, Teresina 64.049-550, Brazil; michelleomaia@ufpi.edu.br; 4Department of Animal and Dairy Science, University of Wisconsin-Madison, Madison, WI 53705, USA; araujorodrig@wisc.edu

**Keywords:** coccidia, natural anticoccidials, sustainable livestock, biomarkers, eimeriosis

## Abstract

**Simple Summary:**

Sheep farming faces major challenges from intestinal parasites called coccidia, causing diarrhea, poor growth, and economic losses. Chemical treatments are becoming less effective due to resistance, while consumers increasingly seek alternatives with less reliance on synthetic compounds in animal husbandry. This study tested whether pods from a Brazilian tree called faveira could naturally control these parasites in lambs. Seventeen young sheep were divided into three groups: untreated, lambs receiving faveira pods as roughage replacement (30.0% of total diet), and chemically treated. Over 45 days, researchers monitored parasite levels and animal behavior. Faveira pods reduced parasite loads by 8.5% compared to untreated animals, while chemical treatment achieved a 36.6% reduction. Notably, animals with higher parasite burdens consumed more water, providing an early warning system for farmers. Faveira pods also decreased environmental contamination by reducing parasite egg shedding by 43.4%. Faveira pods may serve as a complementary approach for parasite management, supporting animal welfare and environmental health in sheep production.

**Abstract:**

Coccidiosis represents a major constraint to sheep productivity worldwide, with increasing concerns regarding anticoccidial resistance and growing interest in reducing dependency on conventional synthetic anticoccidials. This investigation evaluated the anticoccidial properties of faveira pods (*Parkia platycephala* pod—PpP) and their influence on productive performance in naturally infected lambs. Eighteen uncastrated Dorper × Santa Inês crossbred males (20.0 ± 2.5 kg, 5 months) were randomly allocated to three groups: G1 (0% PpP; *n* = 6), G2 (100% PpP replacing roughage, 30.0% of total diet; *n* = 6), and the control group (0% PpP plus 20 mg/kg toltrazuril; *n* = 5). Parasitological assessments, productive performance, and behavioral parameters were monitored over 45 days using oocyst counts, morphometric analysis, digestibility trials, and biometric measurements. Nine *Eimeria* species were identified, with *E. crandallis*, *E. parva*, and *E. bakuensis* representing 53.5% of total oocyst shedding. Group G2 demonstrated a numerical 8.5% reduction in parasite load compared to G1 (*p* = 0.42), while toltrazuril achieved 36.6% efficacy (*p* < 0.05). Species-specific effects were significant for *E. crandallis*, *E. parva*, and *E. ovinoidalis* (*p* < 0.01). A robust correlation emerged between parasite load and water consumption (r = 0.652, *p* = 0.0045), establishing a novel behavioral biomarker for coccidiosis monitoring. Environmental oocyst elimination decreased by 43.4% in the *P. platycephala* group. These findings demonstrate that PpPs possess moderate anticoccidial properties, offering a sustainable complementary strategy for integrated coccidiosis management while contributing to environmental sustainability.

## 1. Introduction

Coccidiosis, or eimeriosis, is a significant parasitic disease affecting sheep globally, with profound economic implications due to its impact on animal health and productivity [1]. Sheep farming is a crucial component of agribusiness, offering multifaceted benefits across various regions and contributing significantly to rural economies [2,3,4]. The sustained growth of the sector in recent years, driven by the growing demand for sheep products and the introduction of more efficient management technologies, highlights the activity’s productive potential [5,6]. However, despite technological advances and the recognized zootechnical potential of breeds adapted to tropical and subtropical conditions, sheep farming still faces significant challenges that limit its productive efficiency and economic sustainability.

Among the limiting factors for sheep production, gastrointestinal parasites stand out as one of the main obstacles to productivity. Coccidiosis, caused by protozoa of the genus *Eimeria* (phylum Apicomplexa), is one of the diseases with the greatest economic impact [7,8], with pathogenic species distributed globally [9]. These obligate unicellular parasites invade the epithelial cells of the gastrointestinal tract, compromising the integrity of the intestinal mucosa and resulting in malabsorption of nutrients, diarrhea, dehydration, growth retardation, and, in severe cases, high mortality, particularly in young animals [10,11,12].

The prevailing climatic conditions in tropical and subtropical regions, characterized by high temperature and humidity, provide a favorable environment for the sporulation and transmission of *Eimeria* oocysts, exacerbating the challenges of disease control and intensifying associated economic losses [13]. In these regions, ovine coccidiosis has a prevalence ranging from 20.67% to 100% in the Brazilian cerrado and semiarid regions, causing economic losses such as reduced feed conversion efficiency, retarded growth performance, and increased mortality rates [14,15,16]. Crossbred sheep populations, particularly Dorper × Santa Inês combinations, are widely used in tropical production systems and demonstrate variable susceptibility to parasitic infections, with genetic variation in parasite resistance being an important factor for sustainable production under challenging environmental conditions [17,18]. This breed-specific genetic variation, combined with oocyst viability under high temperature and humidity conditions typical of tropical climates, necessitates targeted management strategies that consider both environmental and genetic factors in the sustainable control of coccidiosis [18,19].

Conventional coccidiosis control strategies are predominantly based on the use of chemical anticoccidials and, to a lesser extent, on the implementation of strict sanitary management protocols [20]. However, the indiscriminate use of these drugs has resulted in the increasing development of parasite resistance, significantly reducing the effectiveness of available treatments and increasing production costs [7]. Furthermore, growing regulatory and consumer pressures to reduce dependency on conventional antimicrobials and anticoccidials, combined with concerns about antimicrobial resistance development, has intensified the search for sustainable and economically viable alternatives for parasite control [6]. Due to their widespread distribution, integrated parasite management strategies are advised for effective coccidiosis control in goats [9].

In this context, the exploration of natural resources with antiparasitic properties emerges as a promising strategy for developing more sustainable production systems. *Parkia platycephala* Benth., popularly known as faveira, is a leguminous tree native to the Caatinga biome and is widely distributed in the northeast region of Brazil [21]. This species is notable for its abundant production of pods rich in bioactive compounds, including tannins, flavonoids, and saponins, which have demonstrated antiparasitic activity in preliminary studies [22]. Comprehensive toxicological [23] demonstrated that *P. platycephala* leaf and seed extracts exhibit acceptable safety profiles, with most extracts classified as non-toxic (IC_50_ > 1000 μg/mL) or showing only low toxicity (IC_50_ = 598–826 μg/mL) using the *Artemia salina* bioassay. These findings, combined with previous in vivo studies showing no acute toxicity at 1000 mg/kg body weight [24], support the safety of *P. platycephala* for nutritional applications at the doses employed in livestock feeding.

Recent research has shown that the inclusion of byproducts rich in bioactive compounds in ruminant diets can result in benefits that transcend direct antiparasitic effects, positively influencing production parameters such as nutrient digestibility, feed efficiency, body development, and carcass quality [25,26]. In this context, faveira pods (*Parkia platycephala* Pod–PpP), a readily available byproduct from the Brazilian semiarid region, emerge as a promising sustainable alternative that combines economic viability with nutritional benefits for sheep production systems.

Considering the complexity of the interactions between nutrition, animal health, and productivity, it is imperative to develop studies that assess not only the direct effects on parasite load but also the systemic implications for productive performance, body development, and metabolic efficiency. The identification of behavioral and physiological biomarkers that can serve as early indicators of animal health status represents a significant advance for non-invasive animal health monitoring in intensive production systems.

Therefore, this study aimed to evaluate the productive aspects and parasite dynamics of *Eimeria* spp. in naturally infected sheep fed diets containing PpPs as a replacement for Tifton-85 hay. It also aimed to investigate the correlations between parasitological, productive, biometric, and behavioral parameters, with a view to developing sustainable nutritional strategies for coccidiosis control in sheep farming.

## 2. Materials and Methods

### 2.1. Location and Animals

The experiment was conducted at the Science Center of the Federal University of Maranhão (UFMA), in the Lower Parnaíba Region, located at 03°44′33″ S and 43°21′21″ W, in the municipality of Chapadinha, Maranhão. Parasitological analyses were performed at the Applied Parasitology Laboratory of the same institution.

Seventeen uncastrated male Dorper × Santa Inês crossbred lambs with an average initial weight of 20.0 ± 2.5 kg and an approximate age of 5 months were used. They were raised in a semi-intensive system and naturally infected with *Eimeria* spp.

Before experimental initiation, animals underwent a 7-day adaptation period. During this period, all animals received a standardized control diet (0% PpP composition as described in Appendix A) with roughage–concentrate ratio of 20:80, 16.14% crude protein, and 2.81 Mcal/kg metabolizable energy.

Animals received anthelmintic treatment (5% levamisole hydrochloride; Ripercol^®^, Zoetis, Brazil) and vitamin supplementation (Vit ADE, 2.0 mL/animal, Calbos^®^, Brazil). Fecal samples were collected before and 7 days after anthelmintic administration using the modified McMaster technique [27]. Only animals with zero nematode egg counts were included in the experiment. The adaptation diet contained: dry matter 85.28%, crude protein 16.14%, NDFcp 44.5%, total carbohydrates 71.94%, metabolizable energy 2.81 Mcal/kg (Appendix A). Transition to experimental diets occurred on day 0.

The animals were housed in individual 1.45 m^2^ pens located in a covered and ventilated shed, with restricted access to people and equipped with individual feeders and waterers. Mineral salt and water were available ad libitum throughout the experimental period.

### 2.2. Experimental Design and Dietary Management

The experimental design adopted was completely randomized, consisting of three treatments with unequal replication (G1: *n* = 6; G2: *n* = 6; CG: *n* = 5), using initial weight as a covariate. The unequal sample sizes result from the exclusion of one animal from the control group due to health complications unrelated to the experimental treatments during the adaptation period. The treatments consisted of: Group 1—G1 [0% *Parkia platycephala* Pod (PpP) without anticoccidial; *n* = 6], Group 2—G2 [100% PpP without anticoccidial; *n* = 6], and Control Group—CG [0% PpP + 20 mg/kg of Toltrazuril; *n* = 5]. The animals were randomly assigned to the treatments. The nomenclature ‘100% PpP’ refers to complete replacement (100% substitution) of the roughage source (Tifton-85 hay; *Cynodon* spp.) with *P. platycephala* pods, not to the total diet composition.

Isonitrogenous diets were formulated with a roughage–concentrate ratio of 20:80, meeting nutritional requirements for moderately growing sheep targeting 200 g/day weight gain [28]. The detailed ingredient composition and chemical composition of the experimental diets were adapted from previous studies [25,26] (Appendix A). Diet composition (g/kg DM): G1 (0% PpP) contained Tifton-85 hay (30.0), ground corn (20.0), soybean meal (16.7), wheat bran (31.0), mineral salt (2.0), and limestone (0.3); G2 (100% PpP) contained *P. platycephala* pods (30.0), corn grain (20.0), soybean meal (14.5), wheat bran (33.2), mineral salt (2.0), and limestone (0.3). Chemical composition: G1 presented DM 85.28%, CP 16.14%, NDFcp 44.5%, total carbohydrates 71.94%, ME 2.81 Mcal/kg; G2 presented DM 85.78%, CP 15.78%, NDFcp 27.67%, total carbohydrates 71.58%, ME 3.03 Mcal/kg. Mineral supplementation provided (per kg): Na 147.0 g, Ca 120.0 g, P 87.0 g, S 18.0 g, Zn 3800 mg, Fe 1800 mg, Mn 1300 mg, Cu 590 mg, I 80 mg, Co 40 mg, Cr 20 mg, Se 15 mg. Faveira pods were obtained from native trees, sun-dried, ground, and sieved to uniform 2 mm particle size before incorporation.

The experimental diets were provided ad libitum, adjusted daily to 10% leftovers to measure nutrient intake and digestibility. Feed was provided twice daily (8:00 AM and 4:00 PM). Voluntary intake and selection rate were calculated daily by adjusting the feed supply using the formula: (Previous day’s intake/0.90), ensuring that the animals could ingest and select their diet voluntarily [29]. On experimental days 0, 15, 30, and 45, coproparasitological analyses and animal weights were collected. All fecal samples were immediately coded with random numbers to ensure blinded laboratory analysis. The procedures were performed with the animals fasting from solids for 16 h before the first meal of the day.

### 2.3. Qualitative and Quantitative Analysis of Eimeria Oocysts

Fecal samples were collected directly from the rectal ampulla of the animals, wearing gloves and individually identified, and kept at room temperature for analysis in the Applied Parasitology Laboratory of the Federal University of Maranhão. Oocyst counts per gram of feces (OOPG) were determined using the McMaster method modified by Ueno and Gonçalves [19]. These counts were homogenized in 28 mL of saturated saline solution and analyzed in a McMaster chamber, with counts multiplied by 50. All parasitological analyzes were carried out blindly with the identification of samples carried out only after the laboratory analyzes were completed.

Environmental impact was calculated by multiplying daily OOPG values by estimated daily fecal production, based on body weight using the coefficient of 10 g feces per kg body weight per day [30,31]. Daily body weights were estimated by linear interpolation between initial (D0) and final (D45) measurements. OOPG values for intermediate days were estimated by linear interpolation between collection points (D0, D15, D30, D45). Daily oocyst elimination was calculated as: Oocysts eliminated per day = OOPG × Daily fecal production (g). Total environmental impact was determined by summing daily oocyst elimination over the 45-day experimental period.

For morphological analysis of the oocysts, the samples were treated in a 2.5% potassium dichromate (K_2_Cr_2_O_7_) solution for 7–10 days at 25–28 °C until reaching ~70% sporulation, and then refrigerated at 4 °C for 1–2 months before analysis. Isolation was performed by flotation in modified Sheather solution (specific density 1.20; centrifugation at 225× *g*, 5 min) and microscopic examination for morphological characteristics (shape, size, color, presence or absence of micropyle and sporocyst residues) and measurements (polar; equatorial and morphometric index—MI) using optical microscopy with a 10 × 18 mm graduated eyepiece [32,33,34,35] with ovine-specific validation [33], through discriminant analysis and algorithmic approaches demonstrating reliable differentiation for the core species driving our conclusions (Appendix A.

### 2.4. Assessment of Productive Performance and Digestibility

The animals’ productive performance was assessed by weighing them on days 0, 15, 30, and 45 of the experimental period, always at the same time and after a 16 h fast. Average daily gain (ADG) was calculated using the formula: ADG = [(Current animal weight − Previous animal weight)/Days between two weighings]. The animals’ weight on day 0 was established as the initial weight, while the weight recorded on day 45 was considered the final weight.

Dry matter intake was determined by the difference between the amount fed and the daily leftovers, which were weighed before each morning feeding. Consumption data were expressed in g/animal/day and as a percentage of live weight. Dry matter (DM) and crude protein (CP) intakes were evaluated in g/animal/day and as a percentage of live weight (BW).

Dry matter intake was determined by the difference between the amount fed and the daily leftovers, which were weighed before each morning feeding. Consumption data were expressed in g/animal/day and as a percentage of live weight. Dry matter (DM) and crude protein (CP) intakes were evaluated in g/animal/day and as a percentage of live weight (BW). Digestibility was assessed by total fecal collection using individual urine-tight bags [29], with collections at 8:00 AM and 4:00 PM. Samples were identified, weighed, and stored at −18 °C. Food, leftover, and fecal samples were analyzed, determining the levels of dry matter, organic matter, ash, crude protein, and ether extract (EE), according to AOAC [36]. Neutral detergent fiber (NDF), acid detergent fiber (ADF), cellulose (CEL), and hemicelluloses (HCEL) were determined according to Van Soest [37].

Digestibility coefficients were determined by the conventional in vivo method, considering nutrients ingested and recovered in feces, calculating the digestibility coefficient (DC) by difference using the validated formula described by Senger et al. [38]: DC = [(Nutrient intake in grams − Nutrient excreted in feces in grams) × 100]/[Nutrient intake in grams]. Dry matter digestibility (DMD) and crude protein digestibility (CPD) were calculated using daily intake values (determined by feed offered minus leftovers) and fecal excretion values [obtained from total fecal collection over consecutive 24 h periods during last five days of the trial period (days 41–45)]. All chemical analyses were performed in triplicate following AOAC [36] standardized procedures, with coefficients of variation below 5% for all determinations.

Water consumption was monitored individually through drinkers with volumetric meters for five consecutive days (days 20, 21, 22, 23, and 24 of the experiment), recording daily consumption in liters. These measurements were taken at the same time (8:00 AM) for standardization, serving as a behavioral biomarker [26] for quantitative assessment of *Eimeria* spp. induced stress through automated monitoring of individual water intake patterns.

### 2.5. Biometric and Body Development Assessments

The Body Compactness Index (CCI) was calculated on days 0 and 45 of the experiment, obtained by the ratio of live weight to body length (kg/cm), as validated by Costa Junior et al. [39] for Santa Inês sheep and confirmed by Costa et al. [40] and Freitas et al. [41] for hair sheep breeds and their crossbreeds. This index objectively represents the conformation of the sheep, with higher values indicating a greater proportion of muscle and adipose tissue relative to body length. Body weight measurements were taken on a digital scale, and body length was measured with a flexible tape measure, from the cranial end of the chest to the ischial tuberosity. The CCI results obtained allowed us to assess the compactness of the animals in this study, providing parameters for morphometric comparison between groups and analysis of meat production potential.

The loin eye area (LEA) was assessed by ultrasonography at five different time points: days 0, 11, 22, 33, and 45 of the experiment. Measurements were taken between the 12th and 13th thoracic vertebrae using ultrasound equipment with a 5.0 MHz linear transducer. The region was previously shaved and conductive gel applied to optimize image quality. Areas were calculated using digital planimetry and expressed in cm^2^.

### 2.6. Statistical Analyses

All data were initially recorded in Microsoft Excel 365. OOPG values underwent log_10_ (OOPG + 1) transformation to mitigate variance heterogeneity and normalize distribution. This method is conventionally employed in coccidiosis studies for parasitological data analysis [42]. The method circumvents undefined values with zero oocysts, preserving relative differences and stabilizing variance across oocyst counts. This transformation facilitates the application of parametric statistical tests that assume normality and homoscedasticity. The proportions of each *Eimeria* species were determined based on the OOPG values, and the Morphometric Indexes (MI) of the oocysts were calculated as the length-to-width ratio (polar/equatorial sizes) [30,33].

The original and transformed values were evaluated by one-way analysis of variance (ANOVA) followed by Tukey’s test, using GraphPad Prism 8 software, with a *p* value < 0.05 considered significant. For variables measured over time (OOPG, OOC), mixed linear models for repeated measures were used. For the morphometric characteristics of the oocysts (polar, equatorial measurements, and morphometric index), a type II factorial ANOVA was applied, considering the *Eimeria* species and the experimental group as the main factors.

A regression analysis was conducted to explore the relationship between the *Parkia platycephala* pod percentage and the minimum observed OOPG. Specific regression models were developed to investigate the relationship between parasite load and water consumption, establishing predictive equations.

The Body Compactness Index was analyzed using Analysis of Covariance (ANCOVA) models, using the initial values as covariates and simultaneously controlling for the average parasite load of each animal. This approach allowed us to evaluate the effects of treatments on body compactness, considering initial conditions and parasite stress.

A principal components analysis (PCA) was performed on 19 variables, including dietary and performance metrics, and *Eimeria* spp. prevalence. The variables were standardized by mean and standard deviation according to Mingoti [43]. The criterion for selecting the number of principal components, based on Kaiser [44], considers only components with an eigenvalue ≥ 1.0 as contributing relevant quality information to the original variables.

*Eimeria* spp. oocyst counts per gram of feces (OOPG) were classified into four groups based on quartile analysis of the oocyst count distribution: the first quartile (Q1) corresponded to the low OOPG group, the second quartile (Q2) was classified as medium-low, the third quartile (Q3) as medium-high, and the fourth quartile (Q4) as high OOPG. Specific correlation analyses: Pearson correlation analyses were performed between all parasitological variables (OOPG and nine *Eimeria* species) and production variables (total weight gain, average live weight, average daily gain, dry matter and crude protein digestibility, water intake, body compactness index, and rib eye area), both globally and stratified by experimental group. Variables and principal components were interpreted for correlations, with the degree of separation indicating positive correlation, independence, or negative correlation [45]. Pearson correlation was selected over Spearman due to logarithmic transformation of parasitological variables [log_10_(OOPG + 1)] which normalizes asymmetric distributions. With limited sample sizes (*n* = 5–6/group), Spearman would result in 90% data loss in the control group while Pearson preserves analyzable relationships across multiple species-variable combinations (Appendix A).

Composite efficiency indices were developed to integrate multiple dimensions of animal performance: production efficiency index (relating ADG and digestibility to parasite load), biometric index (normalizing body growth by infection intensity), muscle index (focusing on the relationship between AOL development and parasitism), and global index (integrating multiple production dimensions weighted by the square of the parasite load).

All statistical analyses were performed using R software version 4.4.2 (R Core Team, 2024) in RStudio [46]. For linear mixed model analyses, the packages “lme4” [47], “emmeans” [48] for multiple comparisons and estimated means, and “lmerTest” for significance testing in mixed models were used. Multivariate analyses were conducted using “FactoMineR” [49] for principal component analysis and correspondence analysis, “factoextra” for visualization and interpretation of multivariate analyses, and “missMDA” for imputation of missing data using an iterative PCA algorithm.

For correlation analyses, the packages “corrplot” [50] for correlation matrices and “ggcorrplot” for advanced correlation visualization were used. Data manipulation and visualization were performed using the “readxl” packages for importing data in Excel format, “writexl” for exporting results in Excel format, “dplyr” and “tidyr” for data manipulation and organization, and “ggplot2” for advanced graphical visualizations. Specialized statistical analyses were conducted using “rstatix” for parametric and nonparametric statistical analyses, “scales” for formatting scales in graphs, and “binom” for calculating confidence intervals. For morphometric analysis of oocysts, the “agricolae” packages were additionally used for Tukey’s multiple comparison tests and generating statistical groups, and “multcompView” for formatting and visualizing multiple comparison groups.

A post hoc power analysis was conducted to check and validate the unequal group design using packages pwr for power calculations, car for variance tests, dplyr for data manipulation, and rstatix for additional analyses. The harmonic mean of group sizes (*n* = 5.62) was employed in calculating statistical power via the pwr.anova.test() function, with k = 3 groups, α = 0.05, and observed Cohen’s f effect sizes. The design’s relative efficiency was assessed by contrasting the harmonic mean with the arithmetic mean of group sizes. The robustness of statistical assumptions was confirmed through Levene’s test for variance homogeneity and the max/min group ratio criterion (≤1.5) for ANOVA robustness evaluation.

Graphs illustrating infection dynamics and other relationships were created in GraphPad Prism 8.0. Missing data imputation was performed using the iterative PCA algorithm with three principal components as estimated by the cross-validation criterion.

## 3. Results

### 3.1. Parasite Dynamics of Eimeria spp.

The oocyst count per gram of feces differed significantly between the experimental groups (*p* = 0.0067) (Table 1). Significant main effects were observed for group (F = 6.808; *p* = 0.0067) and time (F = 3.755; *p* = 0.0164), as well as a significant group × time interaction (F = 6.336; *p* < 0.001). The estimated transformed OOPG means [log_10_(OOPG + 1)] were: control group (CG) 1.770 (95% CI: 1.325–2.216), G2 group 2.556 (95% CI: 2.279–2.833) and G1 group 2.792 (95% CI: 2.468–3.116) (Figure 1a). Significant differences were identified between CG vs. G1 (*p* = 0.0106) and CG vs. G2 (*p* = 0.0270), while G1 vs. G2 showed no statistical difference (*p* = 0.4204) (Appendix A). Although not statistically significant, G2 showed a numerical reduction of 8.5% in parasite load compared to G1, while toltrazuril achieved significant reduction of 36.6% (Appendix A).

Group G2 showed numerically lower OOPG values compared to G1 (2.556 vs. 2.792 log_10_ units), representing an 8.5% numerical reduction, although this difference was not statistically significant (*p* = 0.4204). Toltrazuril (CG) achieved a statistically significant 36.6% reduction compared to both G1 (*p* = 0.0106) and G2 (*p* = 0.0270) (Appendix A).

Post hoc power analysis confirmed adequate statistical power for the unequal group design. The configuration (*n* = 6,6,5) demonstrated adequate statistical power (>80%) for biologically relevant effects: DMD (Cohen’s f = 1.097, Power = 95.8%, *p* = 0.004), OOPG (Cohen’s f = 1.629, Power = 100.0%, *p* < 0.001), and AWI (Cohen’s f = 0.711, Power = 64.5%, *p* = 0.057). The design showed 98.5% relative efficiency compared to a perfectly balanced design, representing only 1.5% efficiency loss. Levene’s test confirmed homogeneous variances for primary variables (DMD: *p* = 0.983; OOPG: *p* = 0.534), and the max/min group ratio (1.2) met the robustness criteria for ANOVA

The temporal evolution showed distinct patterns for each group (Table 2 and Figure 2). The control group showed a reduction on day 15 (0.74 Log OOPG), with a subsequent increase on days 30 and 45. Group G1 maintained high parasite loads, reaching a peak on day 45 (3.19 Log OOPG). Group G2 showed a reduction on day 30 (2.12 Log OOPG), followed by an increase on day 45.

Nine *Eimeria* species were identified: *E. crandallis, E. parva, E. bakuensis, E. ashata, E. faurei, E. ovinoidalis, E. granulosa, E. pallida,* and *E. intricata* (Figure 1b and Appendix A). The percentage composition of the species varied between groups and over time, with three dominant species (*E. crandallis*, *E. parva*, *E. bakuensis*) representing 53.5% of the total oocysts shed. Morphometric identification was statistically validated through discriminant analysis of 919 measured oocysts, with morphometric data showing perfect concordance with reference values [33]; (differences <0.4 μm. Hierarchical taxonomic analysis achieved classification accuracy of 86.6% with the three species showing significant treatment responses [*E. crandallis* (92.8%), *E. parva* (93.3%), and *E. ovinoidalis* (78.8%)] achieving high identification reliability, confirming methodological adequacy for the core findings of this study, obtaining a classification accuracy of 86.6% with the three species that showed significant treatment responses: *E. crandallis* (92.8%), *E. parva* (93.3%) and *E. ovinoidalis* (78.8%), achieving high identification reliability and confirming the methodological adequacy for the main findings of this study (Appendix A).

*E. crandallis* showed statistically significant differences between groups (F = 4.593; *p* = 0.033). G1 shed 33.7 ± 20.3 million oocysts, G2 shed 16.7 ± 15.7 million, and CG shed 2.9 ± 1.9 million (Table 3).

While overall parasite load differences between G1 and G2 (Appendix A) were not statistically significant, species-specific analysis revealed significant effects of PpP treatment on *E. crandallis* (*p* = 0.0026), *E. parva* (*p* = 0.0050) and *E. ovinoidalis* (*p* = 0.0073), suggesting selective anticoccidial activity against specific *Eimeria* species.

Species-specific responses to experimental treatments were observed in morphometric measurements (Table 4). Among the eight *Eimeria* species statistically evaluated, two species (25%) showed significant morphometric differences between groups G1, G2, and GC. *E. ashata* showed differences in polar (*p* = 0.006) and equatorial diameters (*p* = 0.009), while *E. crandallis* showed differences in polar diameter (*p* = 0.031) and morphometric index (*p* = 0.003). The remaining six species did not show significant morphometric differences between experimental groups, indicating the species-specific nature of treatment effects. *E. intricata* could not be statistically evaluated due to the absence of data in the control group (GC).

Clinical monitoring was performed throughout the 45-day experimental period and no clinical signs of coccidiosis were observed in any animal, including absence of diarrhea, dehydration, lethargy, or other manifestations typically associated with clinical coccidiosis. All animals maintained normal appetite and behavior patterns.

### 3.2. Productive Performance and Intake—Correlations with Parasite Load

Body weight and growth performance data are detailed in Appendix A and Appendix A. Parasite loads differed among groups (OOPG: *p* < 0.001) with weight gains (TWG: CG = 9.59 ± 2.96 kg, G1 = 10.87 ± 1.99 kg, G2 = 10.97 ± 2.47 kg; *p* = 0.610) and average daily gain (ADG: CG = 192 ± 59 g/day, G1 = 217 ± 40 g/day, G2 = 219 ± 49 g/day; *p* = 0.610) statistically similar across groups. No significant association between parasite load and weight gain was observed (r = 0.299, *p* = 0.243). Significant positive correlations were observed between parasite load and production parameters. In the G1 group, *E. bakuensis* showed a positive correlation with total weight gain and average daily gain (r = 0.924, *p* < 0.01).

The correlations between parasite load and digestibility were species-specific and group-dependent. In the control group, *E. crandallis* correlated positively with dry matter digestibility (r = 0.981, *p* < 0.05) and crude protein (r = 0.963, *p* < 0.05). *E. ashata* showed a negative correlation with crude protein digestibility (r = −0.959, *p* < 0.05). A correlation was observed between parasite load and water consumption (r = 0.652, *p* = 0.0045) (Table 5 and Appendix A). Each unit increase in the logarithm of the oocyst count resulted in a 1.93 L/day increase in water consumption, with a coefficient of determination of 65.17% (Appendix A).

### 3.3. Body Development—Correlations with Parasite Load

The biometric ranking showed G1/G2/CG for most measurements (Appendix A). Rib eye area differed significantly between groups (*p* = 0.0113) with linear growth over the collections (0.605 cm^2^/collection; *p* < 0.001). The observed ranking was G2 > G1 > CG. A significant positive correlation was identified between parasite load and final rib eye area (r = 0.512, *p* = 0.0358) (Appendix A).

### 3.4. Correlation and Multivariate Analyses

The first two principal components explained 57.48% of the total data variance (PC1: 33.7%; PC2: 23.8%). PC1 mainly integrated production variables and parasite load. PC2 was dominated by water intake and digestibility parameters (Figure 3 and Appendix A).

The positioning of the groups in the multivariate space showed a clear separation between treatments, with the control group characterized by lower productivity with greater stability, the G1 group by higher productivity associated with greater parasite stress, and the G2 group by high productivity with intermediate stress.

Group-specific correlations are presented in Figure 4 and Appendix A. Group G1 showed multiple significant correlations: *E. bakuensis* with total weight gain and average daily gain (r = 0.924, *p* < 0.01), *E. granulosa* with crude protein digestibility (r = −0.997, *p* < 0.01), and *E. bakuensis* with dry matter digestibility (r = −0.830, *p* < 0.05). Group G2 showed no statistically significant correlations between parasitological and production variables. The Control Group showed a negative correlation between *E. parva* and average live weight (r = −0.992, *p* < 0.001), in addition to the aforementioned correlations between *E. crandallis* and *E. ashata*.

During the 45 experimental days, 1309.4 million oocysts were eliminated into the environment. Group G1 eliminated 799.6 million oocysts, G2 eliminated 453.0 million and the control eliminated 56.7 million (Table 6; Appendix A). Group G2 showed a 43.4% reduction in oocyst elimination compared to G1 (Appendix A).

## 4. Discussion

### 4.1. Antiparasitic Efficacy of Faveira Pods in Light of the Literature

The results demonstrated that G2 exhibited a numerical decrease of 8.5% in parasite burden relative to G1 (*p* = 0.4204). This observation stands in contrast to the notable 36.6% reduction achieved with toltrazuril. Additionally, species-specific reductions were recorded in the PpP-treated group when compared to the untreated controls, with *E. crandallis*, *E. parva* and *E. ovinoidalis* presented significant decreases (*p* < 0.01) within the treated group. The findings indicate a moderate anticoccidial activity of faveira pods under the assessed experimental conditions.

Tchodo et al. [51] demonstrated that hydroethanolic extracts of *Sarcocephalus latifolius*, *Carica papaya* and *Azadirachta indica* exhibited maximum inhibition of *Eimeria* oocyst sporulation (75.85 ± 1.21%, 74.53 ± 1.65% and 71.58 ± 0.24%, respectively) at concentrations of 75 mg/mL, attributing this activity to the presence of antioxidant compounds such as phenols (56.11 ± 0.33 µg/mL), flavonoids (36.65 ± 1.85 µg/mL), alkaloids, saponins and carbohydrates. Similarly, López et al. [52] reported that methanolic extracts of *Ruta pinnata* exhibited significant anticoccidial activity against oocysts and sporozoites of *Eimeria ninakohlyakimovae*, with time- and concentration-dependent inhibition of sporulation, reaching levels similar to formaldehyde used as a positive control.

The moderate efficacy of faveira pods is paralleled by the findings of Mohammed et al. [53], who demonstrated a dose-dependent anticoccidial effect of the methanolic extract of *Lannea schimperi* leaves against *Eimeria tenella*, with greater activity at a concentration of 100 mg/mL compared to 50 and 25 mg/mL. The authors observed a significant reduction (*p* ≤ 0.05) in the number of schizonts and merozoites in the treated groups, suggesting direct interference with the parasite’s life cycle. Muthamilselvan et al. [54], analyzing 32 plants and 40 phytocompounds with anticoccidial properties, highlighted that the mechanisms of action include interference with the *Eimeria* life cycle, regulation of host immunity, modulation of intestinal bacterial growth, or multiple combined mechanisms.

### 4.2. Bioactive Compounds and Mechanisms of Action

The observed species-specific effects may relate to bioactive compounds documented in *P. platycephala*, including phenolics, terpenes, and fatty acids [23]. Previous studies have shown that plant extracts rich in these compounds can inhibit *Eimeria* sporulation and development [51,54,55,56,57,58,59]. Fernandes et al. [23] chemically characterized *P. platycephala* and identified bioactive compounds, such as phenolics, terpenes, and fatty acids, with immunomodulatory, antimicrobial, anti-inflammatory, and antioxidant activities. These findings provide a chemical basis for the observed anticoccidial effects, considering the mechanisms of action reported for other plants with similar properties.

However, the lack of a significant reduction in parasite load in this study suggests that: (1) the concentrations of bioactive compounds were insufficient at the inclusion level tested, (2) the processing methods may have reduced bioactivity, or (3) the compounds present may not be effective against the specific *Eimeria* species identified. Future studies should include phytochemical characterization to correlate bioactive profiles with biological activity.

### 4.3. Temporal Dynamics and Specificity of Eimeria crandallis

*E. crandallis* exhibited significant intergroup differences (*p* = 0.033) in treatment response. This species constituted 21.3% of total oocyst shedding, representing the principal species influencing environmental outcomes. The ecological pertinence of these findings becomes evident considering the highly pathogenic nature of *E. crandallis* in ovine hosts, which leads to substantial intestinal lesions [60,61].

The observed response of *E. crandallis* to PpP indicates selective action of bioactive compounds, consistent with established literature on natural coccidiosis control. López et al. [52] documented similar selectivity with *Ruta pinnata* against *E. ninakohlyakimovae*, while plant extract studies by Tchodo et al. [51] demonstrated varied responses among different *Eimeria* species. This selectivity pattern may correlate with species-specific gastrointestinal locations and differential vulnerabilities to phenolic compounds.

The importance of species-specific identification, as emphasized by Bangoura & Bardsley [13], reflects distinct pathogenic profiles among *Eimeria* species. Regional variations in species prevalence further support this complexity, with Liu et al. [62] reporting that *E. parva*, *E. ovinoidalis*, and *E. crandallis* exhibit distinct responses to environmental factors. In southern Brazil, both *E. crandallis* and *E. ovinoidalis* were identified as significant causes of severe intestinal damage in lambs [61], corroborating the pathogenic significance observed in the present study

### 4.4. Breakthrough Discovery: Water Consumption as a Biomarker

The robust correlation between parasite load and water consumption (r = 0.652, *p* = 0.0045) represents an original scientific contribution of this study. The predictive model demonstrated that each unit increase in the logarithm of the oocyst count results in a 1.93 L/day increase in water consumption, with a coefficient of determination of 65.17%. This finding establishes water consumption as a sensitive behavioral biomarker of stress caused by *Eimeria* spp., offering a practical tool for health monitoring in intensive individual housing systems.

Behavioral biomarkers represent objectively measurable changes in animal behavior that correlate with specific physiological or pathological states, offering non-invasive monitoring capabilities and opportunities for early detection [63]. The water consumption biomarker identified in this study demonstrates key validation criteria: biological plausibility through strong correlation with parasitic load (r = 0.652, *p* = 0.0045), measurement reproducibility across consecutive monitoring days, and practical applicability for automated health surveillance systems. This behavioral approach provides earlier detection sensitivity compared to conventional clinical assessments, as behavioral modifications typically precede visible pathological manifestations. However, individual water monitoring requires intensive management systems with individual housing. In extensive production systems, the principle could be adapted for herd-level monitoring, where significant increases in total water consumption might indicate emerging parasite pressure.

Furthermore, remote monitoring technologies can detect disease behaviors early after exposure, before impacts on growth, enabling targeted treatment. At the same time, Aboshady et al. [64] highlighted the need for validated biomarkers for parasite resistance, proposing a conceptual model that con-siders factors such as genetic differences, age, and immunological experience.

Water efficiency in livestock has established genetic correlations with production traits [65] in cattle, where water consumption correlated positively with gain and feed conversion (r = 0.68). Wisser et al. [66] and Potopová et al. [67] showed that livestock systems exhibit significant regional variation in water consumption per head, indicating potential for specific biomarkers, while Menendez et al. [68] developed dynamic models demonstrating that changes in nutritional composition result in significant differences in the water footprint (2669 L/kg meat), validating the sensitivity of water intake as a physiological indicator.

The predictive model developed in this study (an increase of 1.93 L/day per logarithmic unit of OOPG, with R^2^ = 65.17%) has significant practical implications for production systems, enabling the implementation of automated monitoring systems that could alert producers to increases in parasite pressure before the appearance of evident clinical signs (details in Appendix A). Although monitoring water usage at an individual level necessitates comprehensive management frameworks, the concept could be adapted for collective monitoring in extensive ovine production, wherein notable escalations in overall flock water intake may signal the onset of parasite challenges.

### 4.5. Parasitism-Production Relationship Observations

A positive correlation between parasite load and muscle development (r = 0.512, *p* = 0.0358) was observed, along with E. bakuensis showing strong positive correlation with weight gain in the G1 group (r = 0.924, *p* < 0.01), reflecting the subclinical nature of infections and contrast with typical expectations for coccidiosis impacts on production parameters. The absence of significant differences in weight gain between groups (*p* = 0.61) despite varying parasite loads suggests that the natural infection intensity remained within subclinical ranges for the animals in this study. These observations highlight the complexity of host–parasite interactions and emphasize the need for careful interpretation of correlation data in parasitological studies. The literature indicates that immune responses to parasites involve multifaceted interactions between immunoglobulin classes and metabolic pathways, influenced by factors such as genetics, age, immune history, and type of infection [64]. Non-invasive biomarkers reveal host molecular adaptations at the cellular level [69], while studies have identified differentially expressed genes linked to infection and protection, illustrating dynamic immune responses with initial suppression followed by compensatory enhancements [70]. This complexity underscores the importance of considering multiple factors when interpreting parasitism-production relationships in experimental settings.

Subclinical coccidiosis has a significant impact on the economics of livestock production [14,71] and the literature has documented the absence of weight loss and high OOPG values as a pattern of this subclinical picture of the infection. High oocyst counts do not necessarily correlate with impaired growth [72], and although coccidial infection can cause reduced weight gain, there is little precise quantification of this relationship. It has been found that OOPG alone does not reliably predict production impact, as clinical manifestation depends on multiple factors, such as immunity and other genetic factors, in addition to parasite load [8,73]. Under natural and experimental conditions, the relationship between body weight and oocyst count in sheep with coccidiosis is often weak or statistically non-existent. Diagnosis and management should consider multiple factors beyond OOPG, and clinical manifestation may be a better predictor of productive impact than simple oocyst count [7,8,73,74].

### 4.6. Modulating Effect vs. Complete Elimination

The absence of significant correlations between parasitological and productive parameters in G2 (100% PpP) contrasted with multiple correlations in G1, suggesting that faveira pods may modulate host–parasite interactions. In the control group, *E. parva* showed a very strong negative correlation with average live weight (r = −0.992, *p* < 0.001), while *E. crandallis* correlated positively with dry matter digestibility (r = 0.981, *p* < 0.05) and crude protein (r = 0.963, *p* < 0.05), demonstrating specific patterns of host–parasite interactions under chemical control. This pattern is corroborated by a study that demonstrated the effective interference of bioactive compounds in the intracellular development of *Eimeria tenella* [75]. However, given the limited overall efficacy and subclinical infection levels, this interpretation requires validation in studies with higher infection pressures.

### 4.7. Multivariate Integration and Environmental Sustainability

The animals that received faveira pod were positioned in an intermediate quadrant in the multivariate analysis (Figure 3), confirming the modulating role of this agro-industrial byproduct in the control of coccidiosis. The “productive axis” (PC1: 33.7%) and the “behavioral-nutritional axis” (PC2: 23.8%) jointly explained 57.48% of the total variance. These results indicate that faveira pod effects transcend parasite control, influencing both behavioral and nutritional aspects in treated animals.

Total environmental impact from coccidiosis was substantial, with 1.31 billion oocysts shed over 45 days by 17 animals. The 43.4% reduction in oocyst shedding observed in G2 compared to G1 represents a significant environmental benefit, validating the importance of effective parasite control strategies. Notably, three species (*E. crandallis, E. parva, E. bakuensis*) were responsible for 53.5% of the total environmental impact, highlighting the concentrated nature of parasitic contamination.

These findings have direct implications for One Health, in which nutritional strategies simultaneously contribute to animal welfare, environmental sustainability, and parasite resistance reduction [76,77], with agro-industrial by-products as sustainable protein sources, maintaining adequate production rates even under high endoparasite loads [78]. These integrated nutritional approaches may represent viable pathways toward sustainable ruminant production systems.

### 4.8. Limitations and Future Perspectives

The unequal replication across experimental groups represents a design limitation that must be acknowledged. However, post hoc analysis demonstrated this configuration was methodologically valid and did not compromise study validity or statistical power. The scientific literature establishes that ANOVA is robust for groups with *n* ≥ 5 and max/min ratio ≤ 1.5, criteria fully met by our design [79,80]. The use of harmonic mean for power calculations in unequal groups is a recognized methodology [81], and our results demonstrate adequate power for detecting biologically significant effects

The study has important limitations that should be considered. Clinical parameters such as diarrhea and other symptoms related to coccidiosis were not systematically evaluated, considering the profile of the subclinical infection detected, which could have added valuable data for the study. The 45-day experimental period, while adequate to evaluate acute effects, may be insufficient to understand the long-term impacts of broad bean pod supplementation. The lack of specific analysis of the bioactive compounds present in the pods used limits the understanding of the mechanisms of action.

Future studies should include detailed phytochemical characterization [38,42], to identify specific active ingredients and optimize use protocols.

### 4.9. Practical and Economic Implications

The results support the technical feasibility of PpP as a component of integrated coccidiosis control strategies. The 8.5% efficacy rate represents approximately one-quarter of the efficacy of toltrazuril. However, this natural resource requires no extraction or purification, being readily available in semiarid regions. Additionally, a significant reduction in environmental impact was observed, with 43.4% fewer oocysts eliminated. This combination positions the legume as a sustainable alternative to the exclusive use of synthetic anticoccidials, especially in semiarid regions where this resource is naturally available.

Practical implementation would require standardized protocols for pod collection, processing, and conservation, as well as inclusion The treatments consisted of: Group 1—G1 [0% *Parkia platycephala* Pod (PpP) without anticocciguidelines tailored to the different animal categories and production systems. The development of monitoring systems based on water consumption could add technological value primarily in intensive individual housing systems, while group-level water monitoring might provide early warning signals in extensive production systems, as suggested by Smith et al. [48] on remote monitoring technologies for early parasitism detection.

## 5. Conclusions

This study evaluated the anticoccidial properties of faveira pods (*Parkia platycephala*) in sheep naturally infected with Eimeria spp. The results demonstrated a numerical 8.5% reduction in total parasite load compared to untreated animals, although this difference was not statistically significant (*p* = 0.42). In contrast, toltrazuril achieved a 36.6% reduction (*p* < 0.05). Notably, species-specific anticoccidial responses were observed, with significant reductions for *E. crandallis*, E. parva, and E. ovinoidalis (*p* < 0.01). The main contributions of this study are the identification of water consumption as a behavioral biomarker correlated with parasite load (r = 0.652, *p* = 0.0045), which can facilitate noninvasive monitoring in intensive production systems, and the demonstration of a 43.4% reduction in environmental oocyst shedding compared to untreated animals. Future research is needed to evaluate different levels of dietary substitution, alternative processing methods, and synergistic combinations to optimize bioactive compound delivery and im-prove efficacy.

## Figures and Tables

**Figure 1 animals-15-02896-f001:**
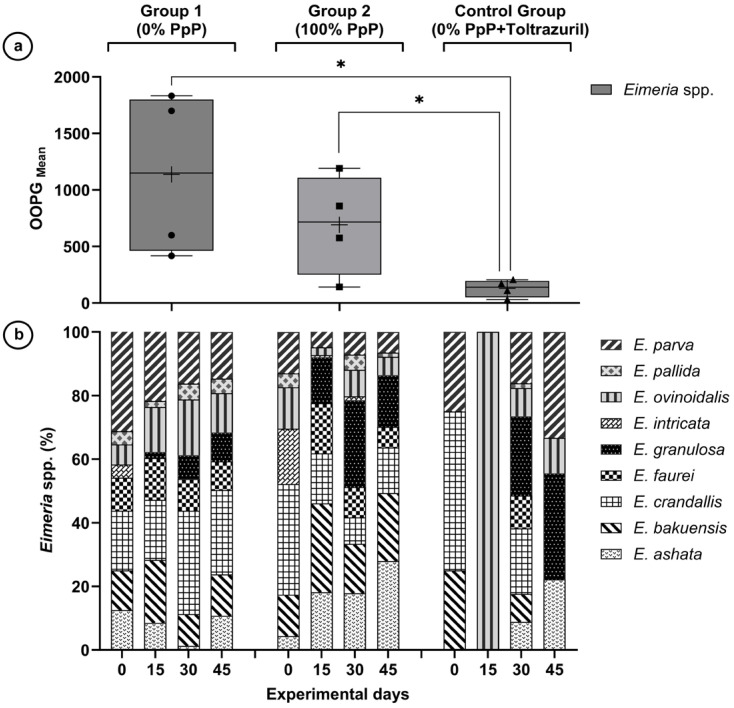
Parasitic dynamics of *Eimeria* spp. in sheep fed *Parkia platycephala* pods. (**a**) Mean oocyst counts per gram of feces (OOPG) by experimental group. Raw OOPG values: Group 1 = 1138 ± 645, Group 2 = 693 ± 450 and Control Group = 130 ± 37. Asterisks indicate significant differences (*p* < 0.05) between groups by Tukey’s test. (**b**) Percentage composition of *Eimeria* species throughout the experimental period (days 0, 15, 30, and 45). G1: 0% *Parkia platycephala* pods without anticoccidial; G2: 100% *Parkia platycephala* pods without anticoccidial; Control Group: 0% *Parkia platycephala* pods + toltrazuril.

**Figure 2 animals-15-02896-f002:**
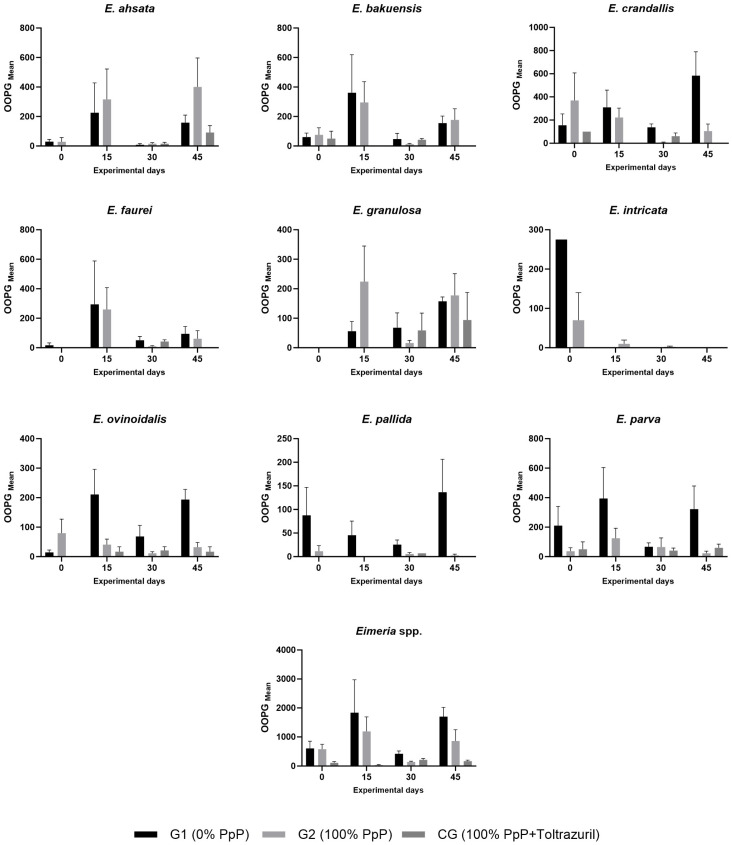
Temporal dynamics of *Eimeria* species by experimental group. Data presented as mean OOPG values for nine identified species plus total *Eimeria* spp. counts throughout experimental days (0, 15, 30, 45). Bars represent mean ± standard error. G1: black (0% PpP); G2: dark gray (100% PpP); CG: light gray (control + toltrazuril). Ten panels show: individual species (*E. ashata*, *E. bakuensis*, *E. crandallis*, *E. faurei*, *E. granulosa*, *E. intricata*, *E. ovinoidalis*, *E. pallida*, *E. parva*) and total *Eimeria* spp. burden. Bars represent mean ± standard error.

**Figure 3 animals-15-02896-f003:**
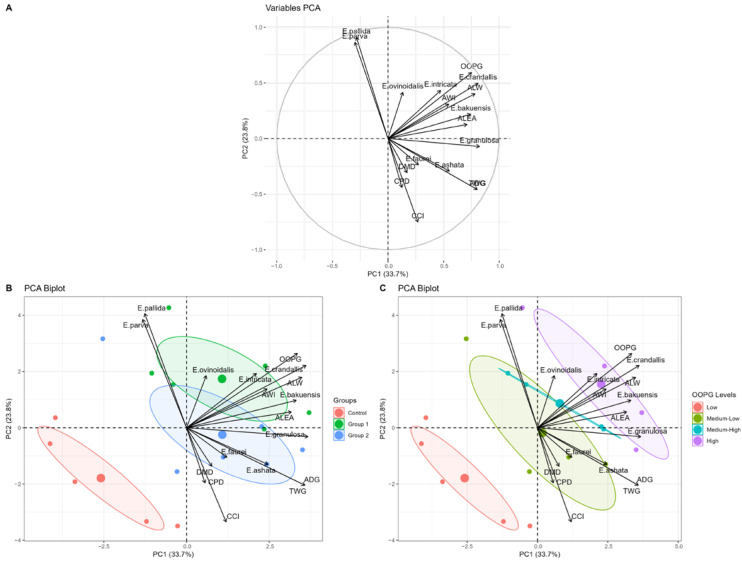
Principal component analysis (PCA) integrating parasitological and productive parameters. (**A**) Correlation circle showing the contribution of each variable to principal components 1 and 2. (**B**) Individual biplot colored by experimental group. (**C**) Individual biplot colored by OOPG levels, evidencing animal separation according to parasitic load. PC1 and PC2 explain 33.7% and 23.8% of total variance, respectively.

**Figure 4 animals-15-02896-f004:**
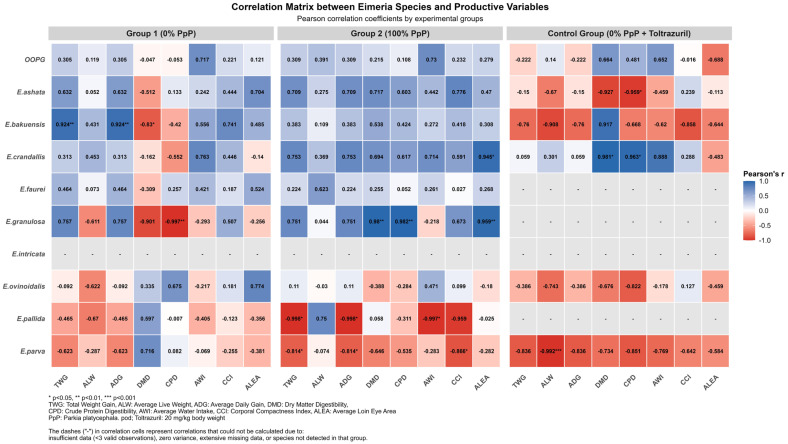
Pearson correlation matrix between *Eimeria* species and productive variables by experimental group. Correlation coefficients represented by color intensity: blue indicates positive correlations, red indicates negative correlations. Asterisks indicate statistical significance: * *p* < 0.05, ** *p* < 0.01, *** *p* < 0.001. TWG: total weight gain; ALW: average live weight; ADG: average daily gain; DMD: dry matter digestibility; CPD: crude protein digestibility; AWI: average water intake; CCI: body compactness index; ALEA: average loin eye area.

**Table 1 animals-15-02896-t001:** Analysis of variance of the linear mixed model for *Eimeria* spp. oocyst counts [log_10_(OOPG + 1)] in sheep fed *Parkia platycephala* pods.

Effect	SS	Num DF	Den DF	MS	F value	*p* Value
Groups	2.83	2	17.0	1.42	6.808	0.0067
Day	2.34	3	51.0	0.78	3.755	0.0164
IW/kg	0.06	1	17.0	0.06	0.266	0.6125
Groups–Day	7.91	6	51.0	1.32	6.336	<0.001

Mixed linear model analysis evaluating the effects of experimental groups, sampling day, initial weight (IW/kg), and their interactions on transformed oocyst counts. SS: sum of squares; DF: degrees of freedom; MS: mean squares. Significance levels: ns = not significant; The significant Groups–Day interaction indicates differential temporal responses between treatments.

**Table 2 animals-15-02896-t002:** Temporal evolution of *Eimeria* spp. oocyst count [log_10_(OOPG + 1)] by experimental group.

Grupo	*n*	Experimental Days				Mean
		0	15	30	45	
G1	6	2.58 ± 0.45 ^b/a^	3.00 ± 0.45 ^b/a^	2.56 ± 0.26 ^b/a^	3.19 ± 0.20 ^a/a^	1.70 ± 0.91 ^NA/a^
G2	6	2.66 ± 0.32 ^ab/a^	2.92 ± 0.40 ^a/a^	2.12 ± 0.18 ^b/a^	2.60 ± 0.65 ^ab/ab^	2.83 ± 0.44 ^NA/a^
CG	5	1.92 ± 0.34 ^ab/a^	0.74 ± 1.02 ^b/b^	1.92 ± 1.08 ^ab/a^	2.20 ± 0.20 ^a/b^	2.58 ± 0.49 ^NA/b^

Values expressed as mean ± standard deviation. Different superscript letters (rows/columns) indicate significant differences (*p* < 0.05) by Tukey’s HSD test. G1: 0% PpP without anticoccidial; G2: 100% PpP without anticoccidial; CG: control group (0% PpP + toltrazuril); NA: Not applicable. The temporal dynamics show that the control group maintained consistently lower oocyst counts throughout the experimental period, while both treatment groups showed variable responses over time.

**Table 3 animals-15-02896-t003:** Analysis of main *Eimeria* species by experimental group.

**Species**	***p* Value**	**Intensity Ranking**	**Interpretation**
*E. crandallis*	0.0026 **	G1 > G2 > CG	Dominant species
*E. parva*	0.0050 **	G1 > G2 > CG	Second most important
*E. ovinoidalis*	0.0073 **	G1 > G2 > CG	Third most important

Statistical analysis of the three most prevalent *Eimeria* species showing significant differences between experimental groups. All three species followed the same pattern with G1 (untreated) > G2 (treated with PpP) > CG (control with toltrazuril), indicating progressive reduction in species-specific intensities with anticoccidial interventions. Significance levels: ** *p* < 0.01.

**Table 4 animals-15-02896-t004:** Morphometric analysis of *Eimeria* oocysts showing experimental group effects within each species.

Species	Dimension	G1	G2	CG	*p*-Value
** *E. ashata* **	Polar	33.02 ± 2.09 ^b^	34.22 ± 2.05 ^a^	32.65 ± 1.14 ^b^	**0.006**
	Equatorial	21.76 ± 1.27 ^b^	22.23 ± 0.52 ^a^	22.18 ± 0.00 ^ab^	**0.009**
	MI	1.52 ± 0.089	1.539 ± 0.085	1.472 ± 0.051	0.094
** *E. bakuensis* **	Polar	32.21 ± 1.21	32.37 ± 1.28	32.04 ± 1.42	0.608
	Equatorial	21.67 ± 1.11	21.63 ± 1.03	22.18 ± 0.00	0.398
	MI	1.49 ± 0.093	1.499 ± 0.084	1.444 ± 0.064	0.238
** *E. crandallis* **	Polar	26.19 ± 1.83 ^b^	26.64 ± 1.76 ^ab^	27.42 ± 2.18 ^a^	**0.031**
	Equatorial	19.23 ± 1.25	19.05 ± 1.34	18.64 ± 1.55	0.216
	MI	1.367 ± 0.123 ^b^	1.404 ± 0.12 ^ab^	1.48 ± 0.17 ^a^	**0.003**
** *E. faurei* **	Polar	31.06 ± 1.95	31.36 ± 1.34	31.33 ± 1.20	0.682
	Equatorial	20.69 ± 1.44	20.83 ± 1.24	20.42 ± 1.20	0.708
	MI	1.509 ± 0.148	1.510 ± 0.104	1.538 ± 0.085	0.849
** *E. granulosa* **	Polar	31.32 ± 2.57	32.14 ± 2.56	30.80 ± 1.88	0.069
	Equatorial	20.64 ± 1.42	20.19 ± 1.53	19.96 ± 1.36	0.287
	MI	1.521 ± 0.126	1.600 ± 0.174	1.550 ± 0.143	0.089
** *E. intricata* **	Polar	44.36 ± 0.00	44.36 ± 5.17	-	N/A *
	Equatorial	30.80 ± 1.74	32.04 ± 4.93	-	N/A *
	MI	1.442 ± 0.082	1.395 ± 0.111	-	N/A *
** *E. ovinoidalis* **	Polar	25.78 ± 2.07	25.82 ± 4.45	24.64 ± 1.32	0.559
	Equatorial	19.76 ± 1.39	20.14 ± 1.42	20.33 ± 1.14	0.367
	MI	1.311 ± 0.135	1.288 ± 0.245	1.217 ± 0.113	0.341
** *E. pallida* **	Polar	17.56 ± 2.18	18.66 ± 1.32	19.71 ± 0.00	0.278
	Equatorial	14.94 ± 1.09	14.79 ± 0.00	14.79 ± 0.00	0.938
	MI	1.174 ± 0.114	1.262 ± 0.089	1.333 ± 0.00	0.089
** *E. parva* **	Polar	21.44 ± 1.60	21.72 ± 1.53	21.69 ± 1.67	0.683
	Equatorial	17.86 ± 1.39	18.35 ± 1.85	18.40 ± 1.27	0.213
	MI	1.205 ± 0.103	1.194 ± 0.140	1.182 ± 0.105	0.735

Values represent means ± standard deviation. Within each species and dimension, different superscript letters and bold *p*-values, indicate significant differences by Tukey (*p* < 0.05). MI: Morphometric Index (Polar/Equatorial ratio). * N/A: not statistically evaluated by ANOVA due to insufficient data in the control group.

**Table 5 animals-15-02896-t005:** Significant correlations between parasitic load and productive parameters.

Variable	Correlation (95% CI)	*p* Value	Interpretation
Average WC (L)	0.6524 (0.2502–0.8626)	0.0045 **	Strong
Final LEA (cm^2^)	0.5116 (0.0411–0.7964)	0.0358 *	Moderate
ADG (kg/day)	0.2992 (−0.2120–0.6818)	0.2434 ns	Very weak
CP Digestibility (%)	−0.0957 (−0.5510–0.4035)	0.7147 ns	Very weak
DM Digestibility (%)	−0.0109 (−0.4890–0.4722)	0.9669 ns	Very weak

Pearson correlation analysis between log-transformed OOPG values and key productive parameters. WC: water consumption; LEA: loin eye area; ADG: average daily gain; CP: crude protein; DM: dry matter. Significance levels: ns = not significant; * *p* < 0.05; ** *p* < 0.01.

**Table 6 animals-15-02896-t006:** Environmental impact of oocyst elimination by experimental group.

Group	*n* Animals	Total Oocysts (Millions)	Fold Change vs. Control	*p* Value
CG	5	56.7	1.0	-
G1	6	799.6	11.7	0.0376 *
G2	6	453.0	6.7	ns
**Total**	**17**	**1309.4**	**-**	**-**

Environmental burden assessment showing total oocyst elimination during the 45-day experimental period. * *p* = 0.0376 (ANOVA); G1 vs. CG: *p* = 0.0299 (Tukey’s test). The untreated group (G1) eliminated 11.7-fold more oocysts than the control group, representing a significant environmental contamination risk. The PpP-treated group (G2) showed intermediate values, suggesting partial efficacy in reducing environmental oocyst load.

## Data Availability

The raw data supporting the conclusions of this article will be made available by the authors on request.

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
