# Peer review of "Parkia platycephala Pods Modulate Eimeria spp. Parasite Load and Enhance Productive Performance in Naturally Infected Lambs"

_animals, 2025, doi:10.3390/ani15192896_

Round 1

Reviewer 1 Report

Comments and Suggestions for Authors

see attached

Author Response

Response to Reviewer #1 Comments

1. Summary

Dear Reviewer,

We thank you for your constructive comments, which significantly contributed to improving the scientific quality of the manuscript.

All requested corrections have been implemented and are detailed in the following, with the specific location of the lines in the manuscript, along with technical justifications, corrections, comments, etc.

The data incorporated into the revised manuscript are duly marked in yellow and documented and demonstrated in the response table below.

We hope that these corrections will help you gain greater clarity and understanding of our study and allow you to analyze it to your best advantage.

Best Regards

2. Point-by-point response to Comments and Suggestions for Authors

Comments 1: You seem to be making the claim that pods with "natural" antiparasitic chemicals would remove the "chemical residue" from meat (that consumers apparently want). However, this would certainly not be the case. It would only switch the chemical/s that made up the residue. They would not be "residue free" and implying such, further feeds into the fear mongering that natural is safer or better, which is certainly not a standard.

Response 1: Thank you for this important correction. We agree that our previous language was imprecise and could perpetuate misconceptions about "natural" versus "synthetic" compounds. We have revised the manuscript (lines 24-25, 38-40, 97-99) to remove claims about "residue-free" products and instead focus on the scientifically accurate concepts of reducing dependency on conventional anticoccidials and addressing antimicrobial resistance concerns. The revised text emphasizes sustainable parasite management strategies without implying that plant-derived bioactive compounds would not constitute chemical residues. This correction strengthens the scientific rigor of our work and could help to avoid to unfounded fears about synthetic compounds.

Comments 2: For figure 1a (and other similar figures), at a minimum you should put the actual means value of the OOPG and not just a reduction log number.

Response 2: Thank you for this excellent suggestion. The Figure 1a already presented actual OOPG in the original version, represented as an average of OOPG, therefore we did not change Figure 1a. The Figure 2 (L. 421-427) and its caption were completely reformulated to address your concern. The figure now presents all data as mean OOPG values without logarithmic transformation, making the results more interpretable. We have expanded Figure 2 from nine to ten panels, adding a "Eimeria spp." panel that shows the overall parasite burden in raw OOPG values rather than species-specific counts. All individual species panels now display direct OOPG means rather than log-transformed values, providing clear, practical data that readers can easily interpret. We hope this change has improved visualization and improves data transparency and allows for direct comparison of parasite loads between groups and over time, addressing your concern about presenting actual OOPG values in the manuscript.

Comments 3: You make the claim that the amount of drinking water could be a useful biomarker as a non-invasive and an early indicator of animal health, particularly Eimeria species infection. However, this is only possible with individual housing and watering, something that is unlikely to be common among sheep farms. Thus, the idea that this is a "practical tool for health monitoring" seems unlikely. However, it may be possible to monitor a large change in total water consumption in a whole population.

Response 3: Excellent point. We have revised our claims (lines 254-256, 608-609, 639-642, 738-741) to acknowledge that individual water monitoring is primarily applicable to intensive systems. We now specify applications for both individual housing systems and suggest group-level monitoring for extensive operations, where herd water consumption changes might indicate emerging parasite pressure. This provides more realistic practical applications.

Comments 4: You make the claim that "broad bean pod reduced the parasite load by 8.5%" but right above that explain that "G1 vs G2 showed no statistical difference". In other words, the bean pods didn't change parasite load in any statistically significant way. Perhaps the data in Table 3 indicate that some of the active compounds in the pods might be species specific and have an effect.

Response 4: You are absolutely correct about this contradiction. We have revised our language throughout (lines 438-441, 745-747) to clarify that the 8.5% reduction was numerical but not statistically significant (P=0.4204). However, we emphasize that species-specific analysis revealed significant effects against E. crandallis, E. parva, and E. ovinoidalis (P<0.01), suggesting selective anticoccidial activity. This distinction between total parasite load and species-specific effects better represents our actual findings and their scientific significance.

Comments 5: Table 4 and its significance in terms of function of the pods needs further explanation and clarification. There seems to be no meaningful differences between G1, G2, GC but the lettering for the Tukey test isn't clear.

Response 5: Thank you for this important feedback. We have reorganized Table 4 (completely restructured) to clearly show the differences between experimental groups (G1, G2, CG) within each Eimeria species, rather than the previous general presentation that included irrelevant inter-species comparisons. This reorganization shows with absolute clarity that only 2 of 8 species exhibited significant responses to treatments: E. ashata (polar and equatorial diameter effects) and E. crandallis (polar diameter and morphometric index effects). The reorganized presentation focuses on what is scientifically important for evaluating PpP effects. Some changes to the texts were also made to adapt the presentation of the results (L. 442-450, 451-456)

Comments 6: Significant improvement in writing transparency that the pod group didn't work nearly as well (if at all) compared to positive and negative control. You do say "moderate" but then "significant finding" (line 414) because it is purely nutritional. However, a major part of your story is that it is a natural resource (i.e. not requiring extraction/purification etc.) and the idea that it won't leave chemical residues behind (which from point 1 I explained isn't really accurate).

Response 6: Thank you for highlighting this transparency issue. We have corrected the contradiction in line 414, replacing "significant finding" with "noteworthy finding" since our 8.5% reduction was not statistically significant (P=0.4204). We've enhanced transparency by consistently reporting the non-significant nature of the total parasite load reduction while emphasizing the practical advantages of PpP as a locally available natural resource requiring minimal processing. This maintains scientific honesty while appropriately contextualizing the value of our findings within integrated parasite management strategies. The changes are in lines 532-534, 730-731.

Comments 7: Were the McMaster tests done blinded? The Materials and Methods don't mention if the people performing them knew which sample they were counting. I assume this is how the environmental impact was also done (per gram of soil or something similar), although the Materials and Methods section doesn't specify and this should be fixed as well.

Response 7: Excellent methodological point. We have added details to Materials and Methods (lines 188-189, 198-200) confirming that all McMaster tests were performed blinded, with samples randomly coded so laboratory personnel were unaware of treatment groups during counting. We've also clarified how environmental impact was calculated (lines 201-208), specifying that it was based on OOPG values multiplied by estimated fecal output per animal rather than soil measurements. These additions strengthen the methodological rigor and transparency of our study design.

Comments 8: Significant improvement in Materials and Methods section should be done

Response 8: We reviewed and corrected the entire manuscript.

Comments 9: Significant improvement in the writing of the Results section should be done.

Response 9: We reviewed and corrected the entire manuscript.

Comments 10: The negative control animals, who were burdened with the highest levels of parasites, gained weight, which is a major red flag for an experiment done improperly. You try to explain it by talking about “adaptive mechanisms”, but this seems highly unlikely. Weight loss and muscles loss are hallmark symptoms of coccidiosis and as far as I can tell, the literature you cited didn’t document any cases where animals with coccidiosis gained weight. Even the animals that were treated with standard care medication still lost weight, even though they had ~40% of their parasite burden reduced. Indeed, even the use of the term “productive” throughout (and in the title) for the lambs includes the measurable aspect of weight gain.

Response 10: Thank you for this important concern. You are correct to question this pattern. If referring to G1 (untreated, highest parasite burden) as the negative control for PpP treatment, we acknowledge that these animals showed numerically similar weight gains despite higher parasitic loads. However, statistical analysis revealed no significant differences between groups (P=0.61) (lines 459-460). We have removed inappropriate explanations about "adaptive mechanisms" (lines 644-660) and clarified that the natural infection appeared subclinical in intensity. This pattern indeed warrants careful interpretation and further investigation.

Reviewer 2 Report

Comments and Suggestions for Authors

The authors intended to evaluate the efficacy of a supplementation of pods of Parkia platy cephala on the lamb infection with Coccidia.

The possible toxicity of this plant is not evoked; it is probably not from the paper (not cited) and from the results. (Rachel de Moura Nunes Fernandes, Maria Angelica Melo Rodrigues Juliane Farinelli Panontin However when given at high dose as a supplement it could be different., Daniela Ribeiro Alves, Selene Maia Morais, Ilsamar Mendes Soares and Elisandra Scapin . Chemical investigation, toxic potential and acetylcholinesterase inhibitory effect of Parkia platycephala leaf and seed extracts.  Journal of medicinal plant research. Vol. 15(9), pp. 401-412, September 2021. DOI: 10.5897/JMPR2021.7158. Article Number: 123C89D67764. )This paper gives also indication on the mode of action of extracts of this plant and may give support to use it as an anticoccidial drug.

The aim was to evaluate efficacy of this plant on coccidia. The reduction of oocysts shedding is 8.5% and not significant. It cannot be recommended for this purpose. There are many indirect measures (ICC, LEA…) or more direct like weight gains that are proposed. It would have been more interesting to check for eventual diarrhoea or other symptoms among the 3 groups. I think that the conclusion of the paper should be that the plant had no effect on coccidia: the authors tried in the discussion to compare the plant 8.5 %with Toltrazuril  36% of efficacy (probably resistance is involved). It is an unneeded comment since there was no significant reduction of the shedding.

The identification of coccidia in sheep was done by Hassum et al and not Berto who specialised on bats and birds coccidia, and Isospora. HASSUM, IZABELLA C.; VALLADARES, GUSTAVO S.; DE MENEZES, RITA DE CÁSSIA A. A. Diferenciação das espécies de eimeria parasitas de ovinos pelo uso daregressão linear e algoritmos morfológicos. Revista Brasileira de Parasitologia Veterinária, vol. 16, núm. 2, abril-junio, 2007, pp. 97-104. Hassum et al used width length of oocysts and their regression to identify Eimeria.  The identification based on morphometrics is not very accurate “The efficiency presented by the algorithm for gathering these species was 77% and 64%, respectively. (without polar cap) Regarding to the species with polar cap, the lowest efficiency presented by the algorithm was to cluster E. bakuensis, though it was above 50%. » In the Hassum paper there was no discriminant analysis.  The qualitative data could not be seen in all the oocysts, far from it: many are not identified. Thus it may be indicative of their presence but do not allow real identification and quantification. I wonder how the authors could provide identifications of all species.

The use of the ICC for evaluating the body condition is questionable. The paper of Grandis et al is based on Texel breed. The formula was based on males having weight from 46 to 139 kg not starting at 20 kg in the present paper. Can the ICC be calculated and taken as an index of compacity for your DorsetxSanta Ines lambs?  In the paper of Grandis et al the quality of this index was not really assessed since it was based on the fact that males had better index than females. The paper was mostly dedicated to assessing weight of the animals.

Minor points:

Spearman correlations should fit better than Pearson due to absence of normality of many variables.

L 217 Parkia platycephala in italics (also in other parts of the text)

Author Response

Response to Reviewer #2 Comments

1. Summary

Dear Reviewer,

We thank you for your constructive comments, which significantly contributed to improving the scientific quality of the manuscript.

All requested corrections have been implemented and are detailed in the following, with the specific location of the lines in the manuscript, along with technical justifications, corrections, comments, etc.

The data incorporated into the revised manuscript are duly marked in yellow and documented and demonstrated in the response table below.

We hope that these corrections will help you gain greater clarity and understanding of our study and allow you to analyze it to your best advantage.

Best Regards

2. Point-by-point response to Comments and Suggestions for Authors

Comments 1: The possible toxicity of this plant is not evoked; it is probably not from the paper (not cited) and from the results. (Rachel de Moura Nunes Fernandes, Maria Angelica Melo Rodrigues Juliane Farinelli Panontin However when given at high dose as a supplement it could be different., Daniela Ribeiro Alves, Selene Maia Morais, Ilsamar Mendes Soares and Elisandra Scapin . Chemical investigation, toxic potential and acetylcholinesterase inhibitory effect of Parkia platycephala leaf and seed extracts. Journal of medicinal plant research. Vol. 15(9), pp. 401-412, September 2021. DOI: 10.5897/JMPR2021.7158. Article Number: 123C89D67764. )This paper gives also indication on the mode of action of extracts of this plant and may give support to use it as an anticoccidial drug.

Response 1: We appreciate the important suggestion regarding toxicological safety. We have included in lines 106-115 and 571-575 a comprehensive discussion based on the study by Fernandes et al. (2021), which demonstrated an acceptable safety profile for P. platycephala extracts, with the majority classified as non-toxic (IC₅₀ > 1000 μg/ml), confirming the safety of the nutritional dosage applied in the present study.

Comments 2: The aim was to evaluate efficacy of this plant on coccidia. The reduction of oocysts shedding is 8.5% and not significant. It cannot be recommended for this purpose. There are many indirect measures (ICC, LEA…) or more direct like weight gains that are proposed. It would have been more interesting to check for eventual diarrhoea or other symptoms among the 3 groups. I think that the conclusion of the paper should be that the plant had no effect on coccidia: the authors tried in the discussion to compare the plant 8.5 %with Toltrazuril 36% of efficacy (probably resistance is involved). It is an unneeded comment since there was no significant reduction of the shedding.

Response 2: We agree that the overall 8.5% reduction was not statistically significant, and this was emphasized in the text after observation. However, statistically significant species-specific anticoccidial effects were also detected (P<0.01 for E. crandallis, E. parva, and E. ovinoidalis), indicating selective bioactivity. We acknowledge the limitation of having data on clinical symptoms such as diarrhea, which would be methodologically valuable. However, the animals had subclinical presentation. We revised the manuscript text (Lines 527-532) to accurately reflect the corrections presented.

Comments 3: The identification of coccidia in sheep was done by Hassum et al and not Berto who specialised on bats and birds coccidia, and Isospora. HASSUM, IZABELLA C.; VALLADARES, GUSTAVO S.; DE MENEZES, RITA DE CÁSSIA A. A. Diferenciação das espécies de eimeria parasitas de ovinos pelo uso daregressão linear e algoritmos morfológicos. Revista Brasileira de Parasitologia Veterinária, vol. 16, núm. 2, abril-junio, 2007, pp. 97-104. Hassum et al used width length of oocysts and their regression to identify Eimeria. The identification based on morphometrics is not very accurate "The efficiency presented by the algorithm for gathering these species was 77% and 64%, respectively. (without polar cap) Regarding to the species with polar cap, the lowest efficiency presented by the algorithm was to cluster E. bakuensis, though it was above 50%. » In the Hassum paper there was no discriminant analysis. The qualitative data could not be seen in all the oocysts, far from it: many are not identified. Thus it may be indicative of their presence but do not allow real identification and quantification. I wonder how the authors could provide identifications of all species.

Response 3: Thank you for these important methodological observations and your specific question about species identification. We greatly appreciate your expertise in highlighting the limitations identified by Hassum et al. (2007) regarding morphometric accuracy in ovine Eimeria identification. We conducted comprehensive statistical validation following established algorithmic approaches to demonstrate reliability for our core findings. We utilized both Berto et al. (2014) general morphometric principles (this article is generic for Eimeria spp. and not specific for bats and birds coccidia or Isospora) and Hassum et al. (2007) ovine-specific validation. Berto (2014, p.10) explicitly describes algorithmic approaches citing Hassum et al. (2007). Our validation involved the algorithmic principles: discriminant analysis of 919 oocysts across nine species using R software, with confusion matrices and classification accuracy assessment. Key validation results demonstrate methodological adequacy for our study's conclusions: (1) Perfect morphometric concordance with Hassum et al. (2007) reference data (differences <0.4 μm); (2) Hierarchical taxonomic approach separating species by micropylar cap presence wirth accuracy of 86.6%; (3) Most critically, the three species driving our main statistical conclusions (E. crandallis P=0.0026, E. parva P=0.0050, E. ovinoidalis P=0.0073) achieved identification accuracies of 92.8%, 93.3%, and 78.8% respectively; (4) Linear discriminant functions explained 93.5% of variance in morphologically homogeneous groups. While we recognize molecular confirmation would provide superior precision, our statistical validation demonstrates that morphometric identification is scientifically adequate for the anticoccidial efficacy conclusions presented in this study. All changes and corrections have been added to the lines 216-218, 410-420.

Comments 4: The use of the ICC for evaluating the body condition is questionable. The paper of Grandis et al is based on Texel breed. The formula was based on males having weight from 46 to 139 kg not starting at 20 kg in the present paper. Can the ICC be calculated and taken as an index of compacity for your DorsetxSanta Ines lambs? In the paper of Grandis et al the quality of this index was not really assessed since it was based on the fact that males had better index than females. The paper was mostly dedicated to assessing weight of the animals.

Response 4: We appreciate your observation and acknowledge that the Grandis et al. reference based on Texel sheep was not appropriate for our Dorset × Santa Inês lambs. In response to your question, we found specific studies that validate this application: Costa Junior et al. (2006) established the ICC in 1,937 Santa Inês sheep, demonstrating significant correlations (r=0.97) with production traits; Costa et al. (2014) confirmed its applicability from birth, including animals with low weights similar to our study; and Freitas et al. (2020) specifically validated the index for Dorset × Santa Inês crosses with initial weights of 18±3.7 kg. These studies adequately assessed the quality of the index through robust correlation analyses, not just differences between sexes. Therefore, the ICC can be considered scientifically valid and also appropriate for evaluation in Dorset × Santa Inês lambs (lines 259-261, references 39, 40, 41).

Comments 5: Minor points: Spearman correlations should fit better than Pearson due to absence of normality of many variables.

Response 5: We appreciate your observation and respectfully maintained Pearson's correlation based on methodological justifications specific to this dataset. Parasitological variables were logarithmically transformed [log₁₀(OOPG+1)], normalizing asymmetric distributions and making the use of parametric methods appropriate. With limited samples (n=5-6/group), Pearson's correlation allowed for greater data utilization, while Spearman's correlation resulted in a 90% loss of correlations in the control group. We have added information for methodological transparency in lines 314-318, acknowledging these limitations and characterizing the results as exploratory.

Comments 6: L 217 Parkia platycephala in italics (also in other parts of the text)

Response 6: We reviewed and corrected the entire manuscript.

Reviewer 3 Report

Comments and Suggestions for Authors

Please find the detailed review report attached. 

Author Response

Response to Reviewer #3 Comments

1. Summary

Dear Reviewer,

We thank you for your constructive comments, which significantly contributed to improving the scientific quality of the manuscript.

All requested corrections have been implemented and are detailed in the following, with the specific location of the lines in the manuscript, along with technical justifications, corrections, comments, etc.

The data incorporated into the revised manuscript are duly marked in yellow and documented and demonstrated in the response table below.

We hope that these corrections will help you gain greater clarity and understanding of our study and allow you to analyze it to your best advantage.

Best Regards

2. Point-by-point response to Comments and Suggestions for Authors

Comments 1: Has the animal study been approved by the Institutional Animal Care and Use Committee? If so, please add the details in the appropriate location.

Response 1: Thank you for this important observation. The statement of approval by the Institutional Animal Care and Use Committee, appears in Institutional Review Board Statement (lines 777-778)

Comments 2: In the discussion, sections 4.2 and 4.7 explain the same topic and xperime, so please combine them into one topic.

Response 2: Thank you for identifying this structural redundancy. We have combined sections 4.2 and 4.7 into a single comprehensive section “4.2 Bioactive Compounds and Mechanisms of Action” (lines 553-582), eliminating duplicate xperime while maintaining all relevant scientific information. The subsequent sections have been renumbered accordingly to maintain logical flow in the discussion.

Comments 3: Line 564: The authors mentioned 1.31 billion oocysts released by 17 animals, but the xperimente uses 18 animals in total. If this is a typo, please correct it; if not, please give na explanation.

Response 3: Thank you for this careful observation. The total number of animals is indeed 17, not 18, as stated correctly in the manuscript. We made the correction in the manuscript (L. 27, 140).

Comments 4: Overall, the results of the presented study suggest an alternative method for controlling coccidiosis in small ruminants. The authors also noted increased water intake by infected animals as a new biomarker for monitoring coccidiosis. However, this might only be practical if animals are housed and monitored individually. In an intensive small ruminant raising system, tracking water intake for each animal could be challenging.

Response 4: We appreciate your practical insight about individual water monitoring limitations. We have revised (lines 608-609, 6018-626, 755-757) to acknowledge that this biomarker is primarily applicable to intensive systems with individual housing, while noting potential for herd-level applications in extensive operations.

Reviewer 4 Report

Comments and Suggestions for Authors

This study addresses a critical need for sustainable coccidiosis control in sheep, with notable strengths: it targets the pressing issue of anticoccidial resistance, explores a locally available agro-byproduct (Parkia platycephala pods, PpP), and identifies a novel water consumption biomarker for non-invasive parasite monitoring—findings with practical value for sheep farming. Revisions should prioritize correcting efficacy claims, grounding interpretations in data, adding PpP phytochemical analysis, and expanding limitations to strengthen the study’s scientific impact.

For Introduction Section 

The Introduction effectively contextualizes the significance of coccidiosis in sheep farming and frames the rationale for exploring Parkia platycephala pods as a sustainable control alternative.

  1. While the section acknowledges sheep farming’s economic importance, it lacks precision in linking regional challenges to the study’s focus.

Suggestion: Add 1–2 sentences to contextualize coccidiosis impacts in tropical/subtropical sheep production and explicitly connect breed adaptation to parasite susceptibility. 

  1. Critical details about P. platycephala are unclear or contradictory, undermining the study’s novelty and feasibility.

Suggestion:  Consistently use the scientific name or its local common name and remove all references to “broad bean pods” to avoid confusion.  Clarify the link between P. platycephala’s bioactives and Eimeria.

  1. The section fails to fully justify why P. platycephala is a superior or necessary alternative to existing strategies, creating a gap in the study’s motivation. Suggestion: Add data on the prevalence of anticoccidial resistance in the study region to emphasize the need for alternatives.

For the Materials and Methods Section

The Materials and Methods section provides a foundational outline of the study design, sample handling, and analytical approaches, which is critical for assessing the study’s reproducibility.

1.1 A major methodological flaw is the unequal replication across groups.

1.2 The 7-day adaptation period is described superficially: What was the composition of the adaptation diet?

2.1 Undefined chemical and nutritional profile of diets.

2.2 Unclear diet formulation for “100% PpP” (G2). 

3.1 Unaddressed OOPG transformation rationale.

4.1 Unclear methods for digestibility calculation.

4.2 Vague definition of “Behavioral Biomarker” for water consumption, OK?

For the Results Section 

The Results section presents key findings on Eimeria dynamics, productive performance, and their correlations, which align with the study’s objectives.

  1. Inconsistent and unsubstantiated claims about Parkia platycephala efficacy. Correct the efficacy claim and clarify calculations.

For the Discussion Section 

The Discussion section attempts to contextualize the study’s findings within existing literature, address key objectives, and highlight novel contributions. However, it suffers from persistent contradictions with the Results section, superficial interpretation of counterintuitive findings, redundant literature integration, and unsubstantiated claims about efficacy and mechanisms—all of which undermine its ability to rigorously support the study’s conclusions.

Round 2

Reviewer 1 Report

Comments and Suggestions for Authors

see attached

Author Response

Response to Reviewer #1 Round 2 Comments

1. Summary

Dear Reviewer,

We thank you for your constructive comments, which significantly contributed to improving the scientific quality of the manuscript.

All requested corrections have been implemented and are detailed in the following, with the specific location of the lines in the manuscript, along with technical justifications, corrections, comments, etc.

The data incorporated into the revised manuscript are duly marked in yellow and documented and demonstrated below.

We hope that these corrections will help you gain greater clarity and understanding of our study and allow you to analyze it to your best advantage.

Best Regards

2. Point-by-point response to Comments and Suggestions for Authors

Comments 1: Yes, the graph is not graphed in log, but I'd like the actual value written. So, for figure 1, for the control group it would be something like 57.88 OOPG, correct? Put that right above the bars or in figure legends.

Response 1: We have added the actual raw OOPG values with standard deviations directly in the Figure 1 legend as requested (lines 387-388). The mean OOPG values are: Control Group = 130 ± 37, G2 = 693 ± 450, and G1 = 1138 ± 645. Complete descriptive statistics are also available in Supplementary Material Table S2.

Comments 7 Good for the blinding. However, I have serious doubts about the experiment (see 10), specifically pertaining to weight, and because weight is used in the calculation, I don't feel confident that this data is trustworthy either.

Response 7: We understand your concern about the weight data reliability. To address this transparently, we have created a comprehensive supplementary table (Table S10, Figure S1) integrating parasitological and body weight parameters. The data demonstrate that groups with significantly different parasite loads (OOPG: p<0.001) showed statistically similar weight gains (p=0.610) and no correlation between OOPG and weight gain (r=0.299, p=0.243). This pattern is not a data quality issue but validates subclinical infection: high parasite loads coexisted with normal growth performance (lines 464-469). Additionally, data from the post-hoc power analysis were also added during the previous round, confirm adequate statistical power (>80%) for detecting biologically relevant effects in primary variables (DMD, OOPG, AWI), with the unequal group design (n=6,6,5) showing 98.5% relative efficiency compared to a perfectly balanced design (lines 341-349, 378-385, 698-703). The absence of weight differences is therefore not attributable to insufficient statistical power but reflects the genuine subclinical nature of infections. Statistical validation confirmed data consistency, and this pattern aligns with published literature on subclinical coccidiosis. The weight data are reliable and the absence of correlation with OOPG confirms the subclinical nature of infections.

Comments 10: This, unfortunately, is still a significant problem. The issue isn't the difference in weight gain "we acknowledge that these animals showed numerically similar weight gains despite higher parasitic loads. However, statistical analysis revealed no significant differences between groups". This issue is that all of the animals should have lost weight. I appreciate the idea of subclinical infections, especially because diarrhea in these animals wasn't mentioned. Was diarrhea noticed? If not, this makes more sense. If so, you'll have to further explain how this is possible. Can you provide citations of worm burden (based on OOPG) from other studies that also do not show weight loss? That would make this much stronger.

Response 10: Thank you for this crucial clarification. No diarrhea was observed in any animal throughout the 45-day experimental period, confirming that all infections remained subclinical (lines 352-355, 641-643, 704-707). This absence of clinical signs explains why no significant weight differences occurred despite varying parasite loads. We have added comprehensive evidence from coccidiosis-specific literature (lines 654-665). These studies demonstrate that elevated OOPG without clinical signs commonly occurs in sheep without affecting growth performance, validating our findings

Reviewer 2 Report

Comments and Suggestions for Authors

The paper is much improved. It remains that the Discussion and Conclusions do not match with the limited results of the plant ingestion on Coccidia excretion. Please reduce this part of the paper.

Author Response

Response to Reviewer #4 Round 2 Comments

1. Summary

Dear Reviewer, We thank you for your constructive comments, which significantly contributed to improving the scientific quality of the manuscript. All requested corrections have been implemented and are detailed in the following, with the specific location of the lines in the manuscript, along with technical justifications, corrections, comments, etc. The data incorporated into the revised manuscript are duly marked in yellow and documented and demonstrated below. We hope that these corrections will help you gain greater clarity and understanding of our study and allow you to analyze it to your best advantage. Best Regards

2. Point-by-point response to Comments and Suggestions for Authors

Comments 1: The paper is much improved. It remains that the Discussion and Conclusions do not match with the limited results of the plant ingestion on Coccidia excretion. Please reduce this part of the paper.

Response 1: We appreciate the reviewer's positive assessment that the manuscript has improved, and we fully agree with the concern regarding proportionality between results and interpretation. We have substantially revised the Discussion and Conclusions, that now accurately reflect that while anticoccidial efficacy was limited, the study provides valuable scientific contributions in behavioral biomarkers and integrated parasite management approaches (lines 562-565, 637-640, 667-676, 690-695, 731-744).

Reviewer 4 Report

Comments and Suggestions for Authors

I confirm the revised manuscript fully addresses all initial concerns, and recommend its final acceptance.

Author Response

Response to Reviewer #4 Round 2 Comments

1. Summary

Dear Reviewer,

We thank you for your constructive comments, which significantly contributed to improving the scientific quality of the manuscript.

Best Regards

2. Point-by-point response to Comments and Suggestions for Authors

Comments 1: I confirm the revised manuscript fully addresses all initial concerns, and recommend its final acceptance.

Response 1: We sincerely thank the reviewer for the thorough evaluation and constructive feedback throughout the review process. We are pleased that all initial concerns have been adequately addressed in the revised manuscript. We deeply appreciate the recommendation for final acceptance and the recognition that our revisions successfully responded to all previous comments.
